# Optimal Choices in Decision Supporting System for Network Reservoir Operation

**Rapeepat Techarungruengsakul** [1], **Ratsuda Ngamsert** [1], **Teerawat Thongwan** [1], **Rattana Hormwichian** [1], **Kittiwet Kuntiyawichai** [2] , **Seyed Mohammad Ashrafi** [3] **and Anongrit Kangrang** [1,*]

[1] Faculty of Engineering, Mahasarakham University, Kantharawichai District, Maha Sarakham 44150, Thailand
[2] Faculty of Engineering, Khon Kaen University, Khon Kaen 40002, Thailand
[3] Department of Civil Engineering, Faculty of Civil Engineering and Architecture, Shahid Chamran University of Ahvaz, Ahvaz 83151-61355, Iran
[*] Correspondence: anongrit.k@msu.ac.th; Tel.: +66-89-843-0017

**Abstract:** The aim of this research was to identify optimal choices in decision support systems for network reservoirs by using optimal rule curves under four scenarios related to water scarcity and overflow situations. These scenarios were normal water shortage, high water shortage, normal overflow and high overflow situations. The application of various optimization techniques, including Harris Hawks Optimization (HHO), Genetic Algorithm (GA), Wind-Driven Optimization (WDO) and the Marine Predator Algorithm (MPA), in conjunction with a reservoir simulation model, was conducted to produce alternative choices, leading to suitable decision-making options. The Bhumibol and Sirikit reservoirs, situated in Thailand, were selected as the case study for the network reservoir system. The objective functions for the search procedure were the minimal average water shortage per year, the minimal maximum water shortage and the minimal average water spill per year in relation to the main purpose of the reservoir system using the release criteria of the standard operating policy (SOP) and the hedging rule (HR). The best options of each scenario were chosen from 152 options of feasible solutions. The obtained results from the assessment of the effectiveness of alternative choices showed that the best option for normal water scarcity was the rule curve with the objective function of minimal average water shortage per year, using HR and recommended SOP for operation, whereas the best option for high-water shortage situation was the rule curves with objective function of minimal of maximum water shortage using HR and recommended HR for operation. For overflow situation, the best option for normal overflow situation was the rule curves with objective function of minimal average water spill per year using HR and the recommended SOP for operation, whereas the best option for the high overflow situation was the rule curve with the objective function of minimal average water spill per year using HR and the recommended HR for operation. When using the best curves according to the situation, this would result in a minimum water shortage of 153.789 MCM/year, the lowest maximum water shortage of 1338.00 MCM/year, minimum overflow of 978.404 MCM/year and the lowest maximum overflow of 7214.00 MCM/year. Finally, the obtained findings from this study would offer reliability and resiliency information for decision making in reservoir operation for the multi-reservoir system in the upper region of Thailand.

**Keywords:** decision support system; reservoir rule curves; optimization techniques; reservoir operation; standard operating policy; hedging rule

## 1. Introduction

Water resources are finite resources that are essential for human survival. There are now difficulties with water availability, both in flood and drought scenarios [1,2]. Reservoirs were built to retain excess water that might be used in downstream areas. Based on the reservoir design, the reservoir's role is to handle rainwater during the rainy season to prevent floods, and to release water according to the demands for water at the basin's

bottom during the dry season. A multi-purpose reservoir is one that may be used for a variety of purposes, including irrigation, water supply for human consumption and industry, hydroelectric power generation, flood or drought relief, water transportation and downstream water control [3,4]. Whereas managing a unified reservoir is comparable to managing a multipurpose reservoir, there are a few additional factors to consider [5]. However, dealing with both types of reservoirs within a single reservoir is not as challenging as dealing with two reservoirs that function at the same time. These reservoirs are connected and use the same system as the others. This is difficult to manage as a high-performance system [6–8].

To fully utilize the reservoir, efficient reservoir management is essential. Therefore, it can probably be said that the network reservoir is a good water management approach and it is now widely used in many countries [9–11]. The advantage is that they provide reservoir managers with more options in terms of flood and drought prevention [12,13]. In particular, in Thailand, a network of reservoirs ranging from small to large reservoirs, such as the Bhumibol and Sirikit reservoirs, was built in order to meet the high water demand of the Chao Phraya River Basin. The maintenance and operation of reservoirs are a major concern in Thailand [14,15].

Recently, it has been discovered that a new water resource development project has a low probability of success due to its high cost and environmental effect. As a result, the management of existing water resources was adjusted in order to achieve more efficient operations [16,17]. The decision support system (DSS) is one of the solutions for the management of existing reservoirs that have been actioned and is highly beneficial at the moment. It is a technological and management integration approach, in which the obtained information can help us to understand problems and decide on management interventions [18–20]. It is a system that works well with unstable or semi-structured issues [21]. Because the water resource system is complicated, the factors and coherence of two reservoirs must be considered simultaneously, or the effects will be more widespread than with a single reservoir [22]. Decision support systems, which provide a breakthrough in particular for network reservoir management, are commonly used in water resource management and its interactions, such as developing a web-based open-source decision support system. DSS was developed to facilitate real-world engagement with dam-operating agencies to generate the daily optimized release decisions, resulting in significant additional hydropower benefits without compromising other objectives when compared to the conventional operation [23]. As a consequence, a decision support system is a useful tool for applying options to a given circumstance by assessing early data to aid reservoir management decision making [24,25].

Reservoir operation is characterized by the water release criteria and modeling methods for simulating a reservoir system, such as the hydropower rule, pack rule and space rule, as well as changing reservoir characteristics. In prior research, reservoir releases were frequently modeled using the most prevalent reservoir operating standards and guidelines [26]. This is the criterion that enables a huge volume of water to be released in order to meet the downstream water demand [27,28]. On the other hand, under an acute water shortage, it may not be appropriate for the reservoir with a limited amount of inflow, while a considerable downstream water demand is generated. As a result, it is unable to fulfill the demand, leading to a high water deficit [29,30], which may require the use of additional reservoir operating criteria to control the operational situation.

The hedging rule (HR) is a release criterion that delivers water at certain intervals in order to retain water for the following period [31,32]. The hedging rule may be described as an attempt to mitigate severe dehydration that may occur in the future by distributing water more evenly now [33]. It is especially well suited to systems with high water demands, but it has problems with data variations in the amount of water flowing into the reservoir [34]. It becomes considerably more challenging when applied to a reservoir that functions as a system to meet the demand for water sharing in conjunction with the reservoir operating rule curves for determining how much water should be allocated [35,36]. These curves are

used in reservoir operation along with the release criterion. Reservoir rule curves, which are made up of upper and lower limits, represent the upper and lower limitations for regulating water release and storage. The use of rule curves in reservoir operations allows for long-term reservoir management. After 4–5 years of use, it is time to look for the best rule curves to improve efficiency [37,38]. Reservoir rule curves have also been investigated in an attempt to find a solution to the common rule curves using a variety of approaches.

Finding the best reservoir rule curve solutions used to be a trial-and-error process [39,40]. Following this, metaheuristic algorithms such as ant colony [41], Tabu search [42], Genetic Programming [43], the Flower Pollination Algorithm (FPA) [44], Wind-Driven Optimization (WDO) [45], Grey Wolf Optimizer (GWO) [46] and the Marine Predator Algorithm (MPA) [47] were used to find the best rule curves. These strategies resulted in the development of new optimum rule curves for dealing with water shortages and surpluses. Some strategies, on the other hand, have their own set of limitations; some cannot converge to the global optimal, while others are too complex and large [48]. Furthermore, some approaches include an overabundance of parameters, making them difficult to employ. As a result, researchers are trying to articulate answers to these problems.

Interestingly, the Harris Hawks Optimization (HHO) method [49], named after the clever bird, the Harris Hawk, has recently been revealed to be another excellent method for obtaining optimal values. In terms of early performance, the birds form a team and utilize assault techniques [50,51]. In comparison to previous swarm-based optimization techniques, HHO does not need prior knowledge of any parameters besides the swarm's initial population. It does not have a derivative equation, and it is also straightforward to use, durable and thorough. The most useful element of HHO is the balance between exploration and exploitation. As the number of HHO iterations increases, so does the exploratory ability.

Establishing rule curves resulting from various water discharge criteria would help network reservoir management to perform better by providing a variety of decision-making options [52,53]. It enables the user or decision maker to adopt these methods in accordance with their engineering preferences, allowing them to more effectively address the issue of a number of options as well as water storage concerns. The optimized rule curves could provide a number of options that can be combined with other water management considerations, such as economics, the environment, society, etc., to produce choices that are comprehensive and successful across all dimensions [54].

It is clear from the aforementioned arguments that maintaining single and networked reservoirs is fraught with issues and challenges. Even managing networked reservoirs differ depending on the region. This makes it feasible to handle reservoir-specific issues as well, whether the right criteria for water release are utilized in accordance with the primary goals of the reservoir building or even establishing reservoir management solutions where all curve rules are reservoir-specific. According to the literature study above, the HHO technique is quite effective when compared to other processes under the same settings, and it is quite helpful when applied to other difficulties. Therefore, the goal of this work is to create alternative decision-making for optimal rule curves in conjunction with the reservoir simulation model system utilizing the Harris Hawks Optimization (HHO) approach, as well as Genetic Algorithm (GA), Wind-Driven Optimization (WDO) and the Marine Predator Algorithm (MPA). Ultimately, this analysis takes into account the Thailand-based Bhumibol and Sirikit reservoirs. In order to find the best rule curves, the hedging rule criteria are used in conjunction with rule curves in reservoir operation. The results of the study are divided into four main sections according to situations in which the reservoir has previously suffered, which could provide options for decision makers to adopt when experiencing such problems again: (1) a normal water scarcity situation, (2) a high water shortage situation, (3) a normal overflow situation, (4) a high overflow situation.

## 2. Materials and Methods

The procedures and techniques used in this study were designed to develop alternative choices in decision support system solutions for determining the network reservoir optimal rule curves by using optimization approaches. The framework of this study consists of the content and operating procedures depicted in Figure 1.

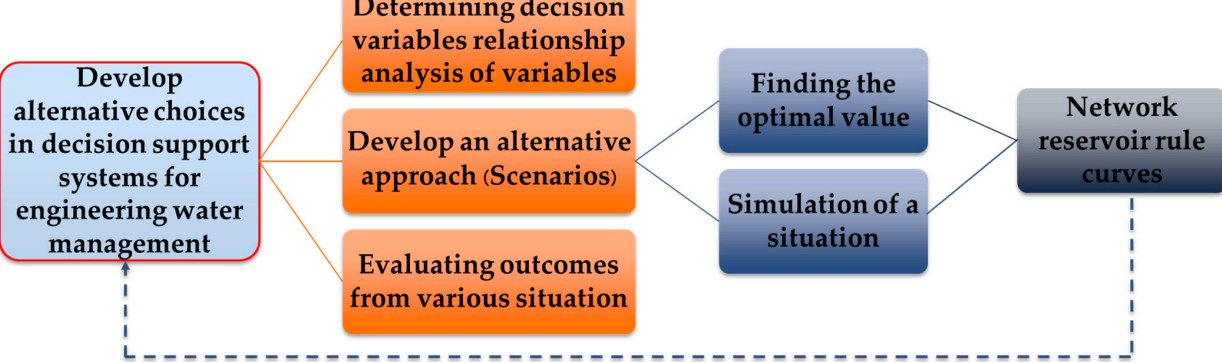

**Figure 1.** Methods of conducting research.

### 2.1. Research Area

The Bhumibol and Sirikit reservoirs, two sizable reservoirs, were selected as the network reservoir system for this study. According to Figure 2, both reservoirs combine their discharge in order to meet the downstream needs. The Greater Chao Phraya Irrigation Project (GCPYIP), the Lower Nan Irrigation Project (LNIP) and the Lower Ping Irrigation Project (LPIP), presented in Figure 2, are the primary irrigation projects that receive water from the two network reservoirs and provide water for downstream uses.

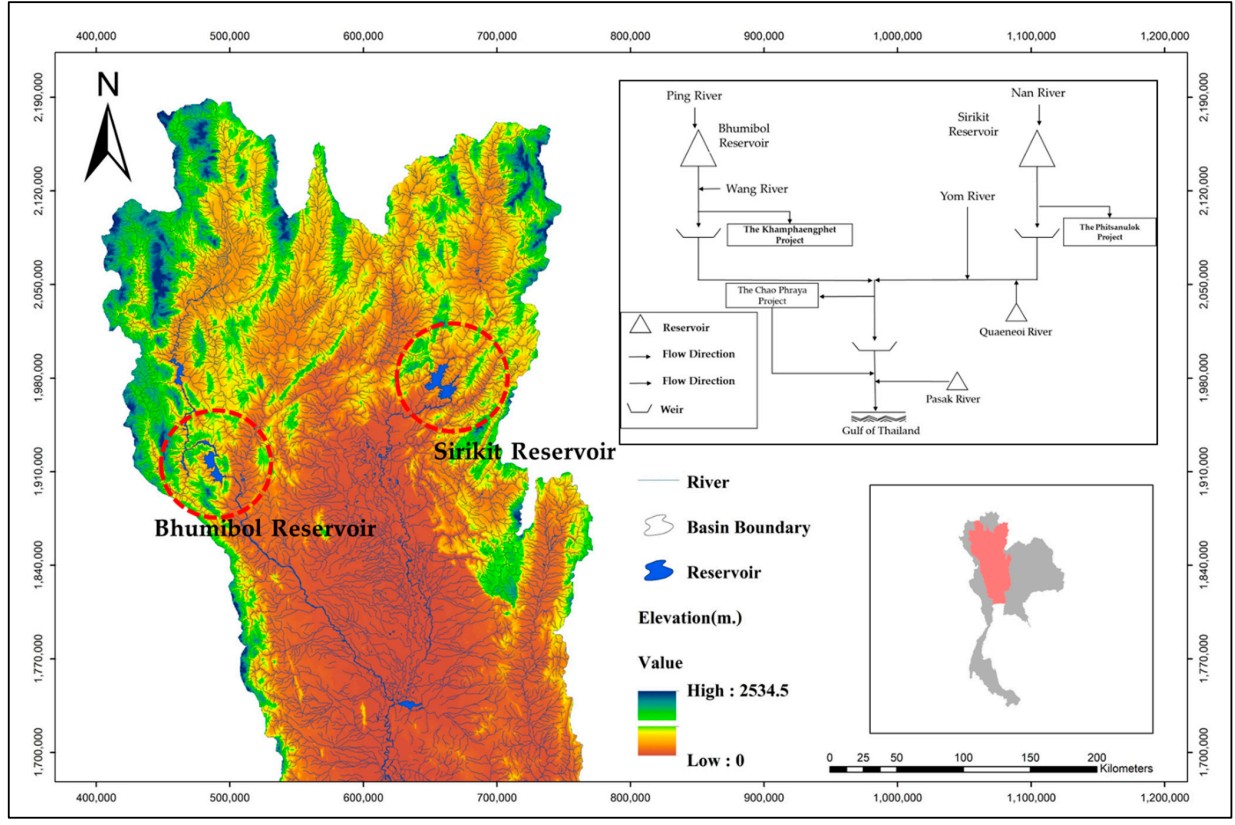

**Figure 2.** Locations and schematic diagram of Bhumibol and Sirikit reservoirs.

The Bhumibol reservoir is located in Tak Province, in the north of Thailand, as shown in Figure 2. It is situated in the upper basin of the Ping River, with the full storage capacity of 13,462 million cubic meter (MCM or $10^6$ m³) and a dead storage capacity of 3800 MCM. The amount of inflow flowing into the Bhumibol reservoir for 57 years (during 1964–2020) is shown in Figure 3.

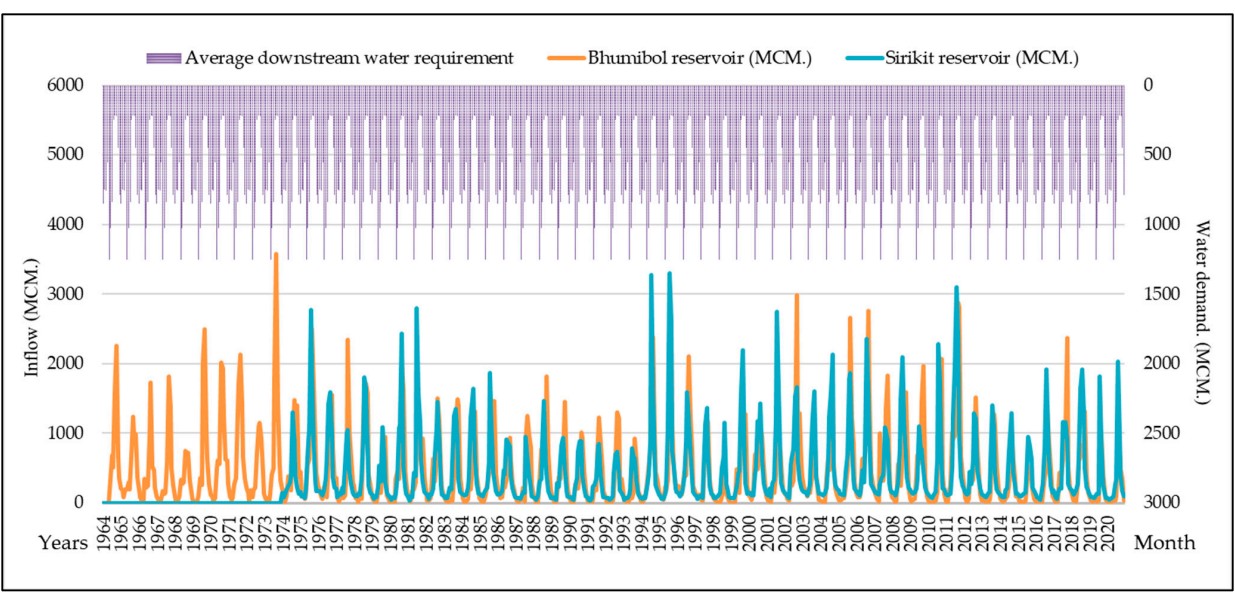

**Figure 3.** Historical inflow and average downstream water requirements of the Bhumibol and Sirikit reservoirs.

The Sirikit reservoir is located in the upper basin of the Nan River, Uttaradit Province, in the north of Thailand (see Figure 2). The full storage capacity is 9510 MCM, and the dead storage capacity is 2850 MCM. The amount of inflow flowing into the Sirikit reservoir for 47 years (during 1974–2020) is also shown in Figure 3.

## 2.2. Establishing a Decision Support System

The data used in this research study were the actual data collected from various scholarly papers, and they were formed as a database. The gathered data were also used for determining parameters to aid in decision making for the design of the effective management of network reservoirs, as seen in Figure 4.

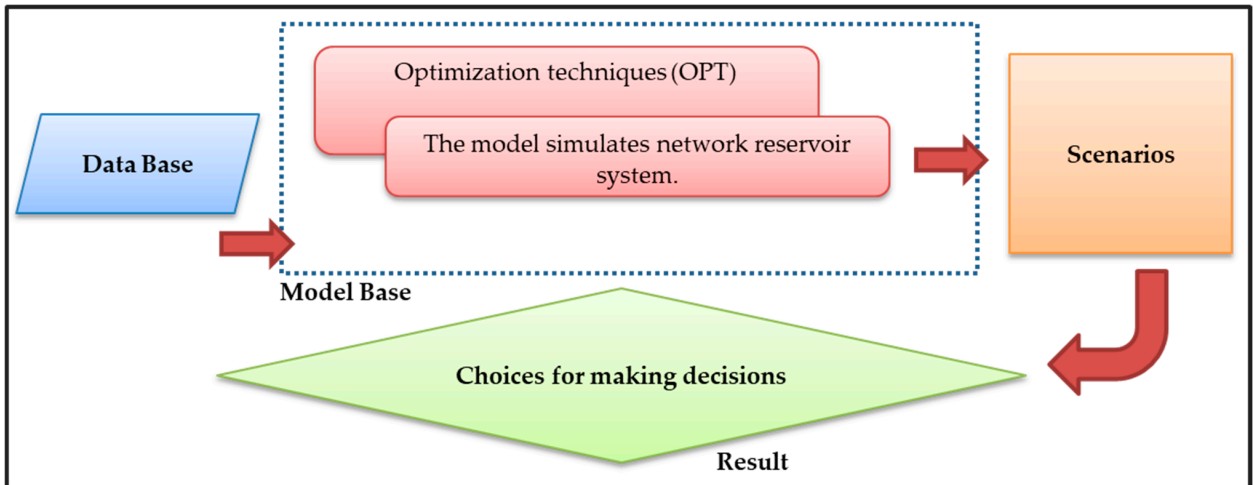

**Figure 4.** The process of making a network reservoir decision support system.

### 2.2.1. Establishing a Database

In determining the proper rule curve for a network reservoir, the database was employed for better decision making in reservoir management. The database layer for this study is depicted in Table 1, and the system simulation model was employed together with four optimization approaches, i.e., the standard operating rule and the hedging rule for network reservoirs, in order to identify the optimal rule curves for operations under different circumstances.

**Table 1.** The database layer for searching the optimal rule curves for network reservoir.

| Data | Period | Source |
|---|---|---|
| Historical inflow | 1964–2020 | Electricity Generating Authority of Thailand (EGAT) |
| Average downstream water requirements | 1964–2020 | Royal Irrigation Department |
| Rainfall data | 1964–2020 | Thai Meteorological Department (TMD) |
| Hydrological data of reservoirs | 2020 | Electricity Generating Authority of Thailand (EGAT) |
| Reservoir physical data | 2020 | Electricity Generating Authority of Thailand (EGAT) |
| Synthetic inflow of 1000 events | 1964–2020 | HEC-4 simulation results |

### 2.2.2. Establishing a Model Base

Network Reservoir Operation Model

The available water, which is determined by the water balance concept, and the water demands from downstream locations compose the reservoir operation system. After taking into account the monthly available water, together with release criteria, operational guidelines and reservoir rule curves, the monthly release of water was projected.

$$W_{v,\tau} = S_{v,\tau-1} + Q_{v,\tau} - R_{v,\tau} - E_\tau \tag{1}$$

where $W_{v,\tau}$ is the available water for month $\tau$ during year $v$; $S_{v,\tau-1}$ is the stored water at the end of month $\tau - 1$ during year $v$; $Q_{v,\tau}$ is the monthly inflow to the reservoir during year $v$; $E_\tau$ is the monthly evaporation loss during year $v$; and $R_{v,\tau}$ is the monthly release during year $v$, which is considered via the operating policy.

Network Reservoir Operating Policy

- Standard operating policy

The reservoir operation model was operated under the standard operating policy, which was created following the concept of the water balance as expressed in Figure 5.

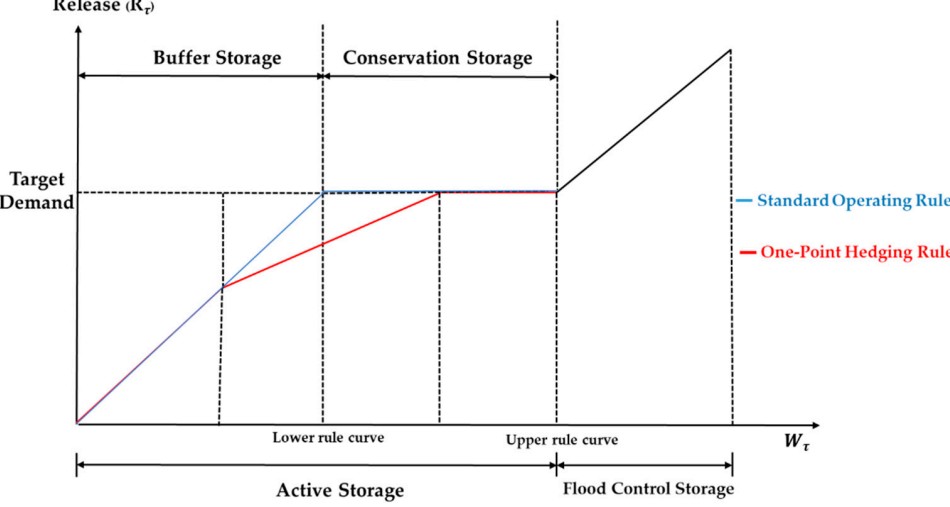

**Figure 5.** Criteria for SOP and HR water release.

● Hedging rule

The hedging rule is a reservoir operating criterion that attempts to reduce the water supply at certain times to store water for later use. Although the amount of water stored in the reservoir can fully meet the target water demand, it can be said that the hedging criterion is an attempt to mitigate serious water shortages that may arise in the future by distributing the water available at present in advance, as expressed in Figure 5.

In this research, a tool was created to determine the proper rule curve for the water storage in the Bhumibol and Sirikit reservoirs as a guideline for releasing water to meet various water demands. The detailed procedures consisted of a reservoir simulation model that simulates water conditions based on water balance principles. Then, the Harris Hawks Optimization (HHO) method was used to find an appropriate rule curve, in comparison to the Genetic Algorithm (GA), Wind-Driven Optimization technique (WDO) and Marine Predator Algorithm (MPA). A better optimization technique will be obtained through Equations (2) and (3) in order to compare outcomes for low- and high-water years in 10-year and 25-year periods, as well as throughout the entire dataset, as can be seen in more detail in Figure 6.

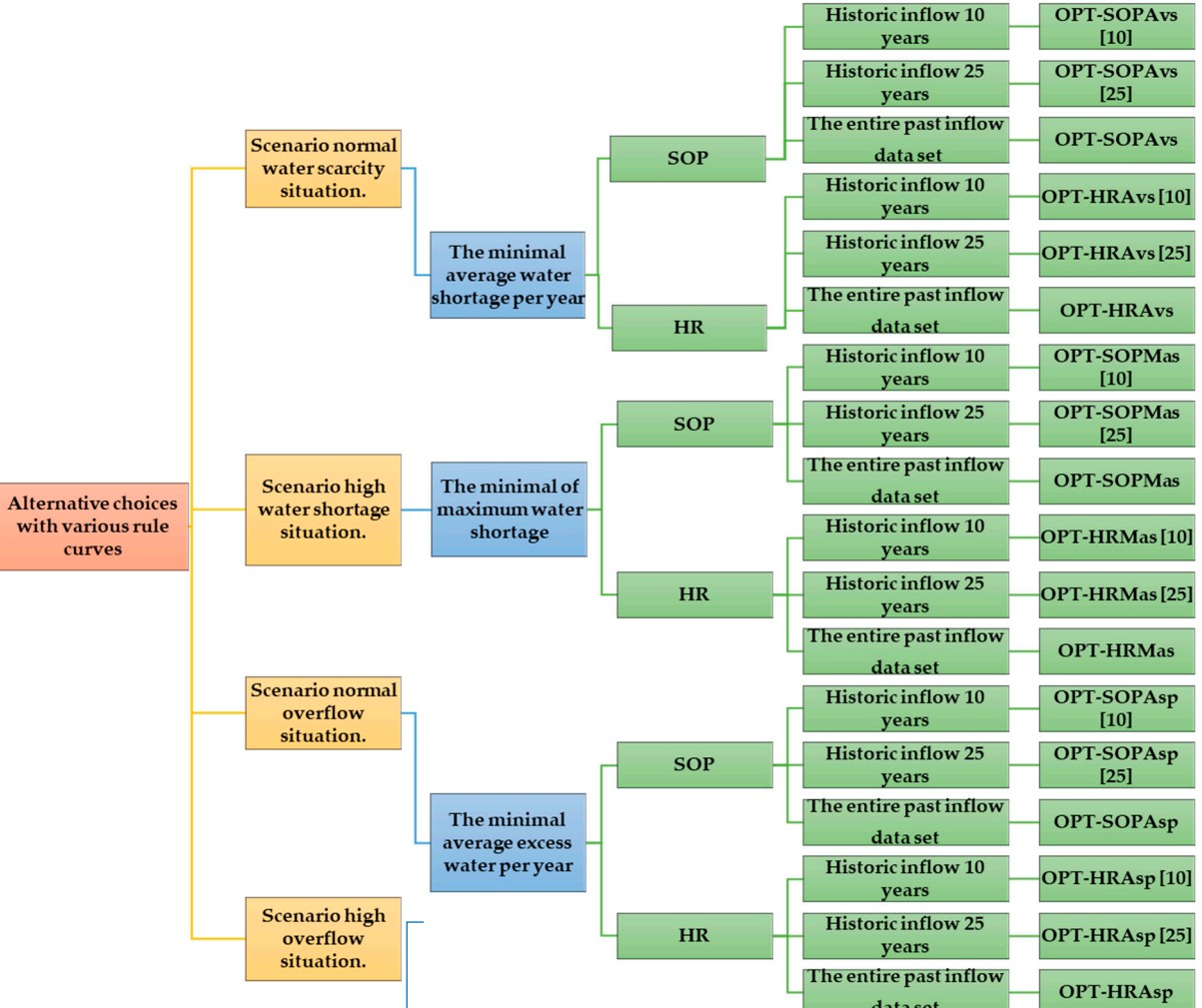

**Figure 6.** Alternative choices for operating network reservoirs under different rule curve scenarios.

The minimal average water shortage per year (Avs):

$$MinH_{(avr)} = \frac{1}{n} \sum_{v=1}^{n} Sh_V \qquad (2)$$

The minimal of maximum water shortage (Mas):

$$MinH_{(Max)} = \sum_{v=1}^{n} Sh_V \qquad (3)$$

The minimal average excess water per year (Asp):

$$Minp_{(avr)} = \frac{1}{n} \sum_{v=1}^{n} Sp_V \qquad (4)$$

here, $n$ is the total number of considered years, $Sh_v$ is the water shortage during year $v$ (the year in which releases are lower than the target demand), and $Sp_v$ is the excess spilled water during year $v$ (the year in which releases are higher than the target demand).

## 3. Results and Discussion

### 3.1. The Best Choice in Decision Making for Network Reservoirs

According to the obtained findings, the decisions for the best set of rule curves were made for each network reservoir decision support system, namely for the Bhumibol and Sirikit reservoirs. The most appropriate decision option was proposed for dealing with water shortage and overflow situations. As shown in detail in Figure 7, there are four application scenarios that may satisfy the decision makers' requirements for reservoir management using the obtained rule curves. Under the SOP and HR criteria, the minimal average water shortage per year, the minimal maximum water shortage and the minimal average excess water per year were the target functions in modeling the network reservoir system.

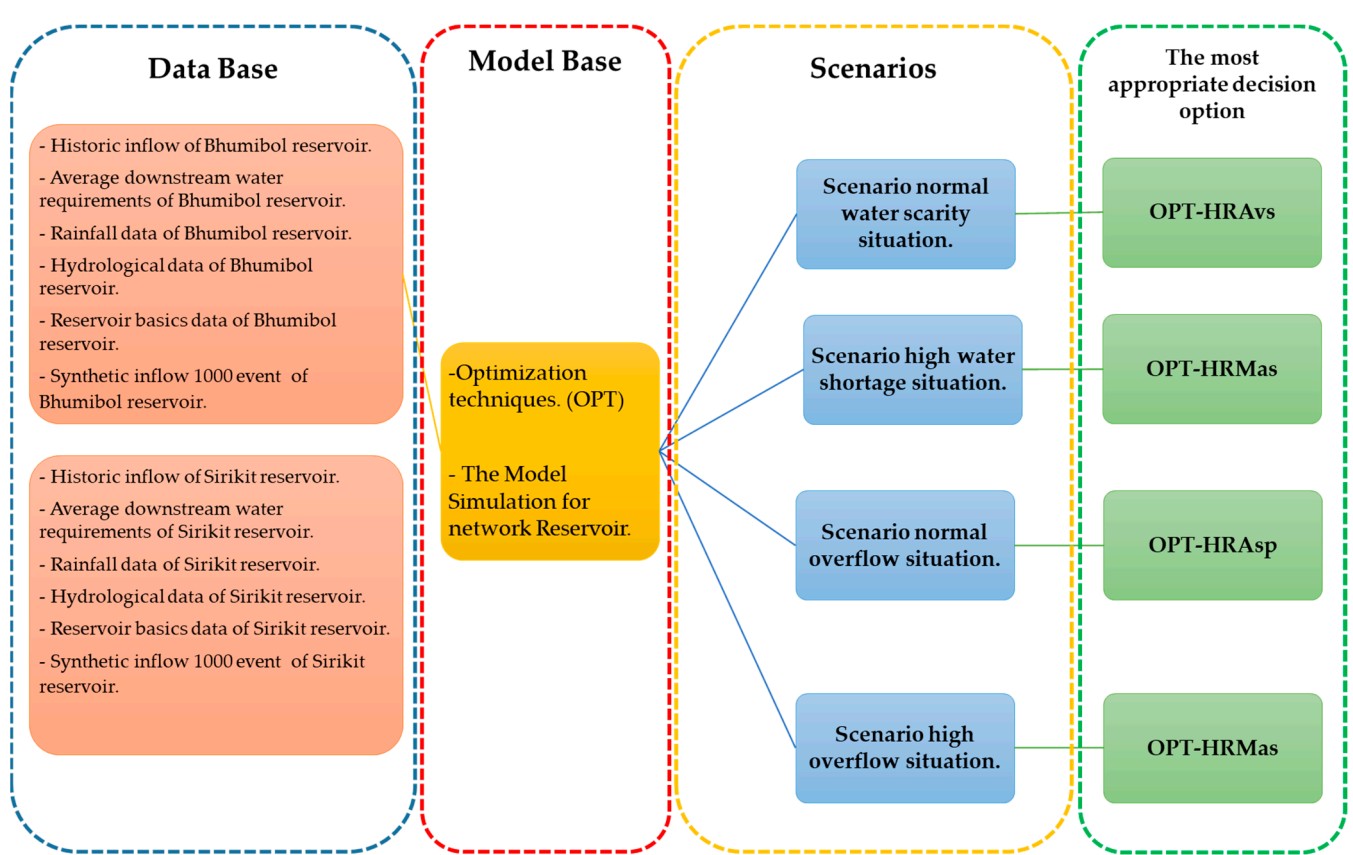

**Figure 7.** The decision-making process for management options of network reservoirs.

The patterns from the new rule curves created by the HHO, GA, WDO and MPA were similar as a result of the same circumstances, and all four approaches could identify the best

solution in each situation to obtain the same solution. The length of the search, however, may vary, and HHO is a new technique that is easy to apply because it balances exploration and exploration. Moreover, HHO can converge to solutions faster than other techniques. In this search, a representative search technique was used, i.e., HHO [55].

Out of 152 possible choices, the four best decisions in the four water situations were taken. In other words, one scenario choice was the best out of 38 possibilities. The rule curves of the four best decisions were chosen to manage the reservoir with the best mitigation approaches for each of the four situations.

### 3.1.1. Scenario of Normal Water Scarcity

The average water shortage for each case of the rule curve are shown in Figure 8. The figure shows the rule curves that a network reservoir should decide to apply when it suffers from a water shortage problem, while a typical shortfall does not have a particularly substantial impact. Clearly, the decision makers must use the rule curves with the objective aim of obtaining the minimal average water shortage per year with the HR criterion (OPT-HRAvs) to release water with an SOP criterion in order to achieve a mean minimum water shortfall of 153.789 MCM/year.

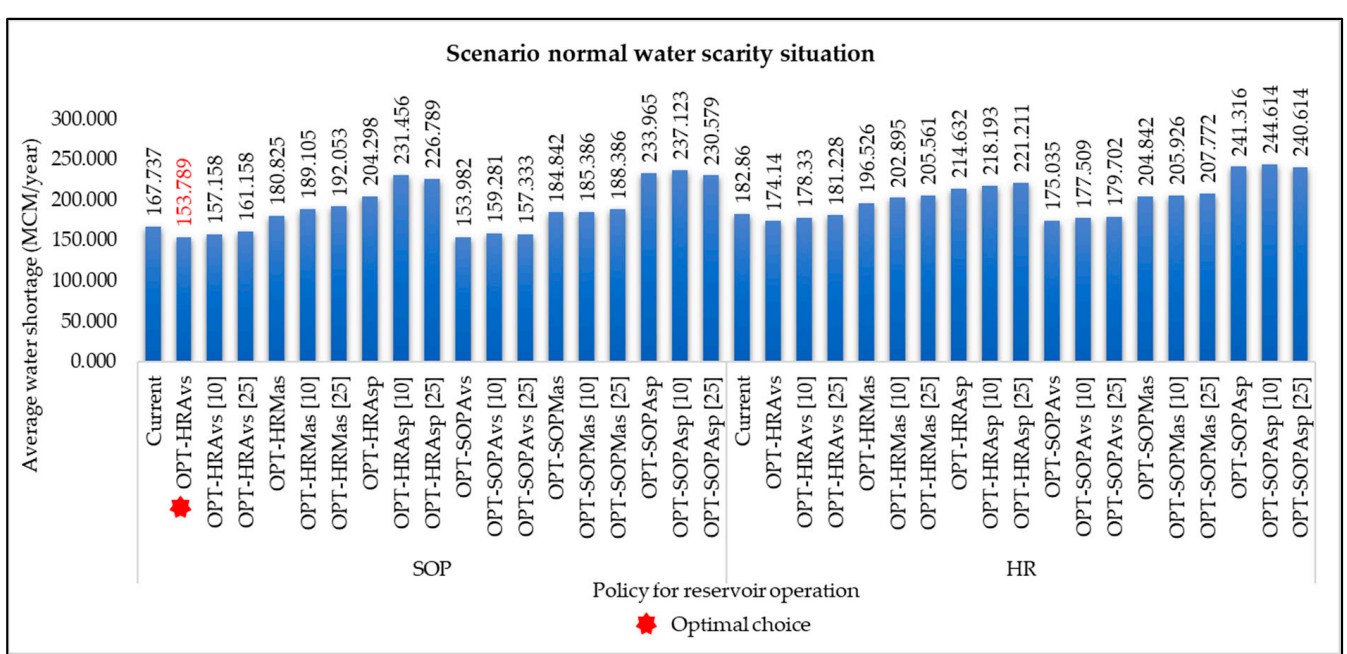

**Figure 8.** Optimal choice for operating rule curves under normal water scarcity scenario.

The results demonstrated that water scarcity could be decreased when the HR water discharge criteria were chosen in conjunction with the objective function. In other words, when the reservoir is not in a critical situation, the SOP criterion should be chosen as the water discharge threshold because the reservoir will continue to release water as much as possible to meet the downstream needs, and this will lessen the reservoir water shortage. Because it will continue to release water as much as possible to meet the downstream reservoir's water needs, this will lessen the reservoir's overall water shortage.

### 3.1.2. Scenario of High Water Shortage

The maximum water shortage values for each rule curve are shown in Figure 9. The figure shows that when network reservoirs experience a water shortage, the adverse impact seems to be very serious. The rule curve that the reservoir should choose to apply is the rule curve with the objective aim of achieving the minimal maximum water shortage with the HR criterion (OPT-HRMas). This means that the decision makers must apply this rule

curve to release water with the HR criterion threshold, which will result in the lowest maximum water shortage of 1338.00 MCM/year.

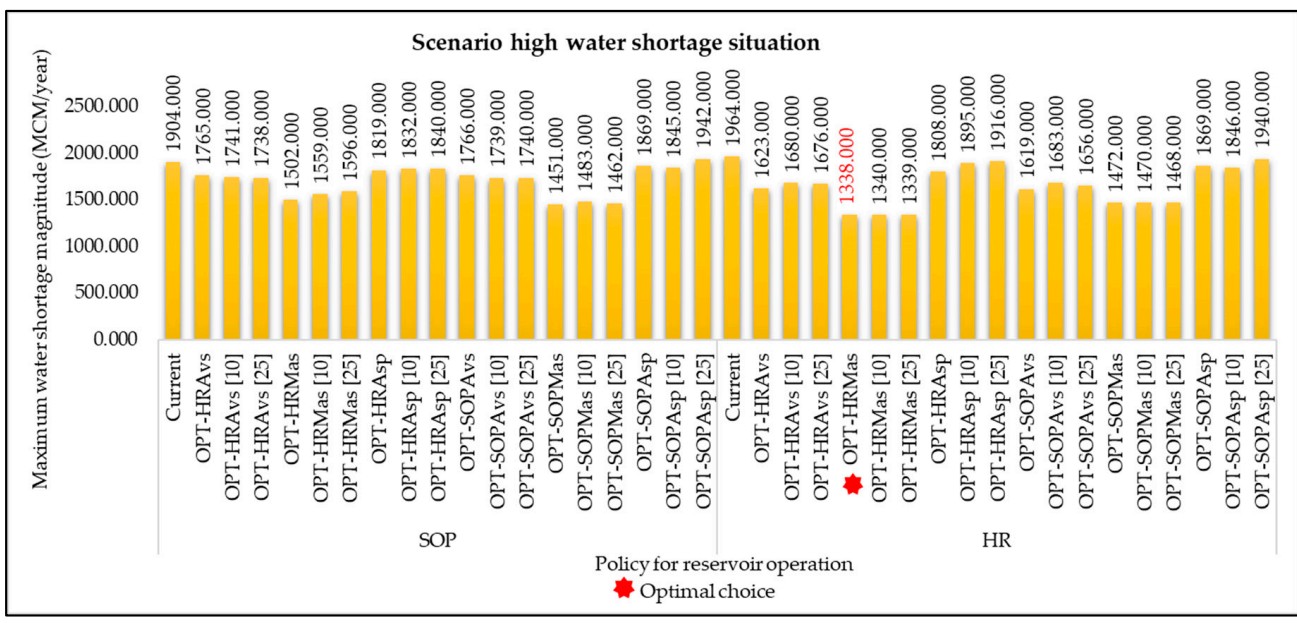

**Figure 9.** Optimal choice for operating rule curves under serious effect of high water shortage situation.

The results showed that when the HR water discharge threshold was selected in combination with the appropriate objective function, a severe peak shortage could be minimized. It can be seen that when the reservoir encounters a severe water shortage situation, the water discharge criterion that the reservoir should choose is the HR criterion. This is because the HR criterion attempts to store water by reducing the release during the dry season, when the reservoir storage capacity is sufficient to meet the water demand. As the main goal of the operating policy, the HR criterion will also release less water than the target requirement in order to avoid a severe water shortage in the next dry season.

### 3.1.3. Scenario of Normal Overflow

Figure 10 illustrates the average excess water for each rule curve. The figure also shows that when the network reservoir experienced an overflow problem, it was merely a normal overflow that did not have a serious impact. The rule curve that the reservoir should choose to apply is the rule curve with the objective aim of obtaining the minimal average excess water per year with the HR criterion (OPT-HRAsp). In this case, the decision maker must use the established rule curves for releasing water with the SOP release threshold, which will result in a mean minimum overflow of 978.404 MCM/year.

Obviously, the results also demonstrated that using the HR water discharge criteria to find the rule curves in conjunction with the appropriate objective function is not only appropriate for severe water shortages but also lowers the overflow. It can be observed that when a reservoir is not suffering from a severe overflow, the SOP criterion should be chosen, even if the rule curve was generated by the HR situation, since the SOP threshold can ensure the water's release throughout the dry season based on the downstream demand. Under this policy, the reservoir can hold more water, while a reduction in overflow can also be achieved.

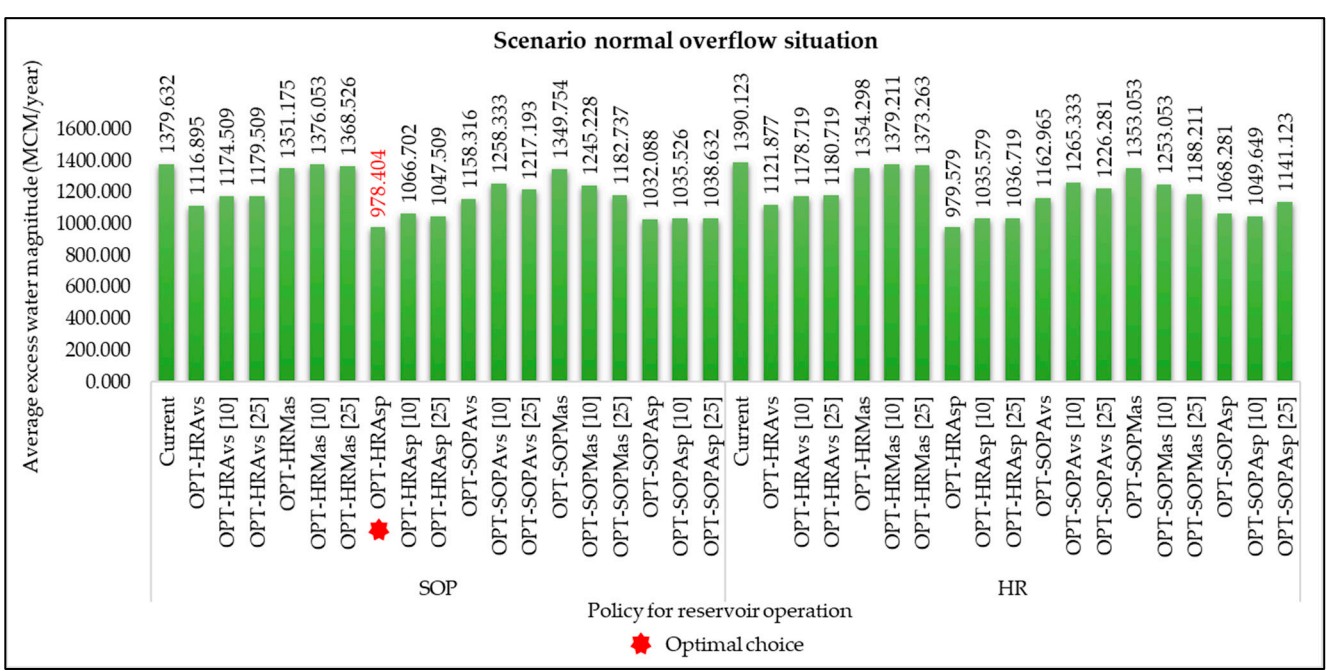

**Figure 10.** Optimal choice for operating rule curves under typical overflow situation.

### 3.1.4. Scenario of High Overflow

Figure 11 illustrates the maximum excess water for each rule curve. This figure also demonstrates that when a network reservoir encounters a water overflow problem, the impact is expected to be very serious. The reservoir should then use the rule curves generated by the objective function of minimal of maximum water shortage with the HR criterion (OPT-HRMas), which produce the highest overflow value. Likewise, the maximum value of minimum overflow when released with the SOP and HR criteria requirements will be 7214.00 MCM/year.

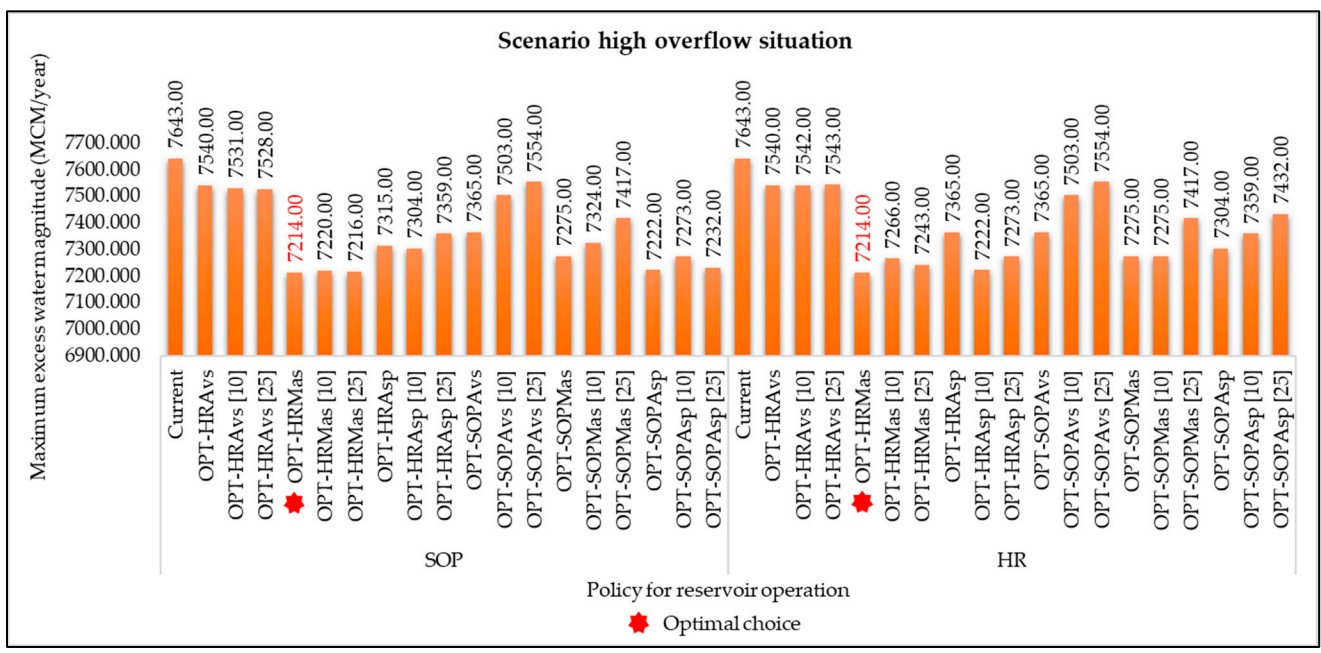

**Figure 11.** Optimal choice for operating rule curves under high overflow scenario.

The results also indicated that the severe peak overflow can be decreased when the HR water discharge criterion is used to determine the rule curve in conjunction with a suitable objective function. It could also be observed that when the reservoir was experiencing a significant overflow, both water discharge criteria could be set. However, when the rule curve generated by the HR threshold was performed together with the HR criterion, the reservoir operation with the rule curve became more efficient and easier to manage.

### 3.2. The Model Basis

The model based within the network reservoir simulation framework consisted of a reservoir simulation model and optimization techniques such as Harris Hawks Optimization (HHO), the Genetic Algorithm (GA), the Wind-Driven Optimization technique (WDO) and the Marine Predator Algorithm (MPA). The structure of the model based within the network reservoir simulation framework is described in Figure 12.

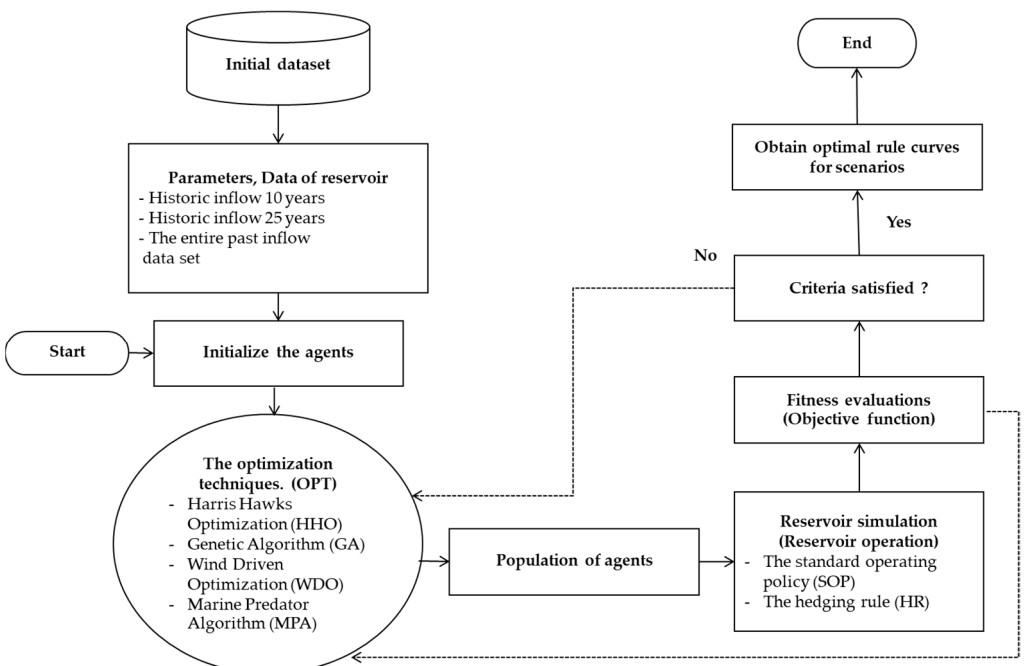

**Figure 12.** The model based within the network reservoir simulation model.

- Firstly, the model starts with the input data and all initial necessary data, such as the upper and lower bound data of the reservoir and objective function.
- The optimization technique starts with a set of network reservoir rule curves generated for the initial population {X1, X2, ..., Xn} that is created randomly within the feasible space (note: the feasible space is the value between the dead storage capacity and the normal high water level of the considered reservoir).
- After the first set in the initial population have been calculated (48 simultaneous decision variables consist of 24 values from the upper rule curves and 24 values from the lower rule curves for both reservoirs), for this study, each decision variable represents the monthly rule curves of the reservoir, which are defined as the upper and lower rule curves of the Bhumibol and Sirikit reservoirs.
- The monthly release of water will be calculated by the reservoir simulation model considering these rule curves (fitness evaluations) in accordance with the criteria.
- Next, the released water is used to determine the objective functions that were described in the previous section's procedure. After this, the reproduction process will create new values of rule curves in the next generation (population of agents). Thereafter, the procedure is repeated until the criteria are satisfied and optimal rule curves are then obtained for different scenarios.

- The obtained rule curves based on the above optimization techniques with the same condition on each scenario will be similar according to the optimal solution, and the obtained rule curves are as shown in Figure 13.

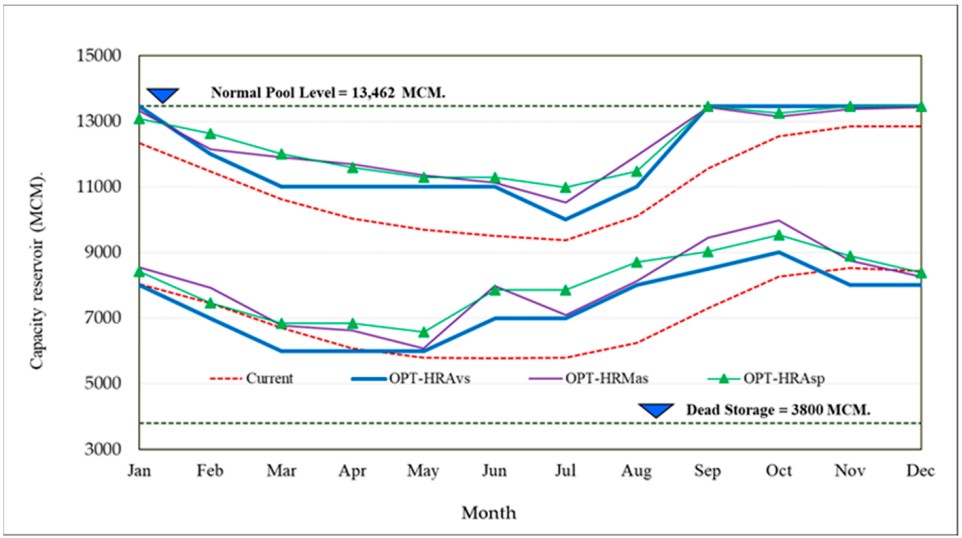

(a) Optimal rule curves of the Bhumibol reservoir.

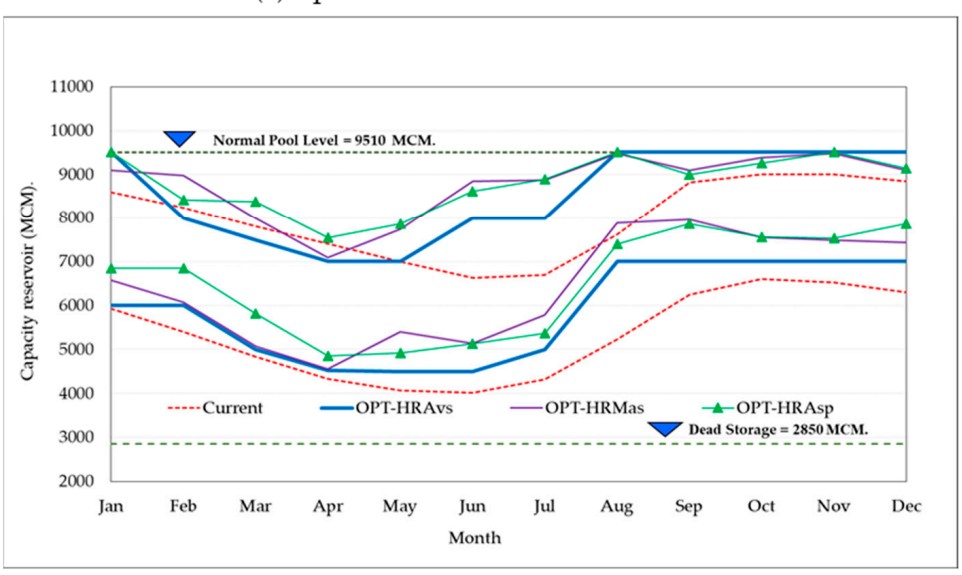

(b) Optimal rule curves of the Sirikit reservoir.

**Figure 13.** The obtained rule curves based on the proposed optimization techniques.

## 4. Conclusions

This study explains the processes of searching for the rule curves that are suitable for network reservoirs and help to identify the best reservoir operation. A decision support system for the network-based reservoir operation of the Bhumibol and Sirikit reservoirs has been developed, which can assist reservoir managers in obtaining guidance, information and scenarios in searching for applicable rules for reservoir operation. Based on this, reservoir operation decision making can be employed to determine the outcome and provide different scenarios that will enable and assist the reservoir managers to monitor any occurrences and serious crises. The obtained system will also enhance the efficiency in the administration of plans based on the demands of executives, in which the system can also play a vital role in defining policies, setting objectives and establishing the organization's vision for success.

Finally, a database system, which can be used as a support tool for analyzing scenarios and identifying and evaluating early responses to arising situations, could be developed. According to the findings of the study, it was found that the Bhumibol and Sirikit reservoirs can be a good case study that provides alternative results for the network reservoir decision support system. In other words, whenever a reservoir water shortage and overflow issue emerge in a variety of scenarios, a decision can be taken to adopt the optimal rule curve in each scenario, in which the four best decisions out of 152 possible choices are available to fulfill the operator demands with adjustable curves for reservoir management. It should be noted that the majority of release criteria were derived from the hedging rule-based rule curves, which can contribute to the alleviation of water scarcity and overflow using both the standard operating policy (SOP) and hedging rule (HR).

The result indicates that the rule curve that a network reservoir should decide to apply when it suffers from a water shortage problem, when a typical shortfall does not have a particularly substantial impact, is the rule curve with the objective function of providing the minimal average water shortage per year with the HR criterion (OPT-HRAvs) to release water with an SOP criterion in order to achieve a mean minimum water shortfall of 153.789 MCM/year. When network reservoirs experience water shortages, the adverse impact seems very serious. The rule curve that the reservoir should choose to apply is the rule curve with the objective function of obtaining the minimal maximum water shortage with the HR criterion (OPT-HRMas), which will result in the lowest maximum water shortage of 1338.00 MCM/year.

In addition, when the network reservoir experiences an overflow problem, and it is merely a normal overflow that does not have a serious impact, the rule curve with the objective function of obtaining the minimal average excess water per year with the HR criterion (OPT-HRAsp) will result in a mean minimum overflow of 978.404 MCM/year. When a network reservoir encounters a water overflow problem, the impact is expected to be very serious. The reservoir should then use the rule curve generated with the objective function of obtaining the minimal maximum overflow with the HR criterion (OPT-HRMas), which will be 7214.00 MCM/year.

The reservoir's purpose searching function must be examined and implemented in the reservoir simulation, as well as searching for a novel approach that is effective, exact and low-complexity for the appropriate rule curves. In the future, they can be applied to other networks.

**Author Contributions:** Conceptualization, R.T. and A.K.; methodology, R.T. and A.K.; validation, R.T. and A.K.; formal analysis, R.T. and A.K.; investigation, R.T. and A.K.; writing—original draft preparation, R.T. and A.K.; writing—review and editing, R.T., R.N., T.T., R.H., K.K., S.M.A. and A.K.; supervision, R.T. and A.K. All authors have read and agreed to the published version of the manuscript.

**Funding:** This research project was financially supported by Mahasarakham University.

**Institutional Review Board Statement:** Not applicable.

**Informed Consent Statement:** Not applicable.

**Data Availability Statement:** This study did not report any data.

**Acknowledgments:** The authors would like to acknowledge the Royal Irrigation Department, EGAT (Thailand), and the Bhumibol and Sirikit reservoirs in the Chao Praya project for supporting this study. The authors would also like to thank the editor and the anonymous reviewers for their comments, which helped in improving the quality of the paper.

**Conflicts of Interest:** The authors declare no conflict of interest.

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
