# Peer review of "Optimal Choices in Decision Supporting System for Network Reservoir Operation"

_water, doi:10.3390/w14244090_

Round 1

Reviewer 1 Report

water-2042839

Title: Optimal Choices in Decision Supporting System for Network Reservoir Operation

1.      The subject and flowcharts are interesting, but some major revisions should be considered. Also, please, start revision in MS-Word> Review menu, while the track change is on for the rapid referee in the next stage. All changes to the manuscript must be indicated by using tracked changes. Require a point-by-point response to the comments, including a description of any additional experiments that were carried out and a detailed rebuttal of any criticisms or requested revisions that you disagreed with.

2.      Please use the following reference(s) in the introduction, and discussion and clarify the differences or consistency between them and the presented paper(s). (A web-based decision support system for smart dam operations using weather forecasts)

3.      Please put some photos of the region, use hillshade in maps, using refer to and cite these: “Spatial Modeling Considering valley's Shape and Rural Satisfaction in Check Dams Site Selection and Water Harvesting in the Watershed” and “Ecotourism and socioeconomic strategies for Khansar River watershed of Iran”

4.      The abstract is not well written. You should include some of the main findings (quantitative) in the abstract section. Add some stronger conclusions in it. The abstract should have a conclusion of the study.

5.      Add more on the basic of the problem in the introduction.

6.      I suggest the author to demonstrate what does the paper add to the current literature? and what new knowledge is added by this study?

7.      It is suggested to present the structure of the article at the end of the introduction. At the end of the introduction add a para including 1-Gaps in the backgrounds you try to fill them, 2-your novelty and unique aspects 3-Hypothesis 4-Objectives.

8.      Discuss the merits and limitations of the technique applied.

9.      Figs’ and Table captions should be more informative text, including the complete abbreviation used.

10.   The presentation fails to discuss the summary, and tries to some vague reason, which is not an explanation. Need to compare the results with new references.

11.   The explanation for the critical analysis is not sufficient, although some of the good points have been identified.

12.   Please rewrite the conclusion with the proper explanation in the R & D and innovation process.

13.   Abbreviations are numerous in this manuscript! They should be explained before the introduction.

14.   The material and method section is too weak in the manuscript and you need to focus on it more. Some abbreviations were not introduced such as HRA, HRM,… in their first appearance.

15.   Please make sure your conclusions section underscores the scientific value added to your paper, and/or the applicability of your findings/results, as indicated previously. Please revise your conclusion part into more detail. Basically, you should enhance your contributions, hypothesis retain/reject, limitations, implications/applications, advantages/disadvantages, policies, underscore the scientific value added to your paper, and/or the applicability of your findings/results and future study in this session.

Author Response

Dear Editor,

We really appreciate the reviewers' comments, which are very detailed and very helpful in improving our manuscript. We have made a major revision to our manuscript. We have improved our manuscript following reviewers' comments. All of the changes have been modified in the revised manuscript. We have addressed all of the comments which are shown below for each one, in which the reviewer’s comments are in black text and authors’ responses are in red text.

Sincerely,

Anongrit Kangrang

Reviewer 2 Report

Based on Harris Hawks optimization algorithm and Hedging rule, In this paper, the optimal rule curves of reservoirs operation are selected under different scenarios, and the research results have certain reference significance for the reservoirs operation management.

The modification suggestions are as follows.

1. It is mentioned that the multi-objective operating of a single reservoir is more complicated than that of multi-stage reservoir, but it is not specified in what aspect is the complexity.

2. Lines 245-249 It is mentioned that the location information of LNIP and LPLP is shown in Figure 2, but I did not find it in Figure 2.

3. In the results and discussion section, it is suggested to add the discussion about HR and SOP, to show how much HR can improve scheduling goals compared with SOP in the same scenarios, and to illustrate the advantages of HR compared with traditional SOP. On the other hand, the current analysis results can only show the difference between the operating objective of HR and SOP. It is suggested to take a certain year as an example to specifically show the operating process of the reservoir in each month, and further demonstrate and analyze the influence of the two rules on the operating process of the reservoir.

Author Response

(The authors gave the same response as above.)

Reviewer 3 Report

The paper “Optimal Choices in Decision Supporting System for Network Reservoir Operation” has several relevant problems that make it unsuitable for publication.

-        The language must be thoroughly revised because some sentences miss the verb, are obscure, or the words used are imprecise. Examples are the title itself (what is a “network reservoir”? Does it stand for “reservoir network”?) or “While maintaining a unified reservoir is similar to managing a multipurpose reservoir, but with a few more aspects to be considered” (line 49) or “At present, it is found that a new water resources development project has a tendency to decline due to its high cost and environmental impact” (line 63) or “four optimization approaches, i.e., the standard operating rule, and the hedging rule” (line 183).

-        Figs 1 and 4 are inconsistent. In fig. 1, some blocks are not connected, and the starting block is already the solution to the problem. In fig. 4, the model that simulates the system has “scenarios” as output instead of being the input of the simulation.

-        The EQs (1) to (3) do not correspond to the definition in the text. For instance, eq. (1) is defined as “The minimal average water shortage per year,” while the formula is simply the average yearly water shortage.

-        The shortage is defined in line 202 as “in which releases are higher than the target demand” instead of the opposite: releases are lower.

-        The simulation models of the two reservoirs are not described. Therefore, it is unclear if there are limits on the released flow or if there is evaporation and seepage and how they are considered.

-        Table 1 shows the data used in the study, while the Data Availability Statement says, “This study did not report any data.”

-        The model base is described after presenting the results.

-        Fig. 11 quotes a “population of agents” not mentioned in the text.

-        Fig. 12 uses “Capacity” instead of “storage” or “volume” and a “normal pool level” (which is again a volume, not a level) not defined in the text.

-        “MCM.” is used instead of the correct M m3 throughout the manuscript.

-        The authors thank “the reviewers” in the acknowledgment, but they probably refer to another journal.

However, these are not the main drawbacks of the paper. The critical problem is that the authors define the management rule as a function of the assumed inflow pattern (see fig. 6) as if the future inflow sequence was perfectly known. Additionally, despite the thousands of papers dealing with the management of multipurpose reservoirs, they judge the system performance under a different objective in each scenario. So, not only the management rule but even the rule objective is a function of the future (unknown in reality) inflow values.
Indeed, the management approach assumed is that of rule curves. Possibly because they are actually in use (however, the “current” values in figs. 7 to 10 are not commented on). Such an old approach refers indeed to a deterministic condition (all inflows are known). In contrast, in a more realistic situation where inflows can only be forecasted with an inevitable error, it is essential to define also what to do above or below the target rule curves.
In summary, the results presented in the paper cannot be used in actual reservoir management; thus, they do not represent a significant addition to the literature.

Author Response

(The authors gave the same response as above.)
